# SELF-SUPERVISED FEATURE RE-REPRESENTATION VIA LENNARD-JONES POTENTIAL LOSS

## ABSTRACT

The Lennard-Jones potential, initially developed to model molecular interactions, is characterized by a repulsive force at short distances to prevent over-clustering and an attractive force at longer distances to maintain balanced proximity, resembling the equilibrium-seeking behavior of particles in natural systems. This offers a potential pathway for more orderly entropy reduction in higher-order features. This paper introduces a self-supervised approach for feature re-representation, utilizing a Lennard-Jones potential loss to constrain the gradient directions between positive and negative features in computer vision tasks. Unlike supervised learning directly driven by downstream tasks or contrastive learning with multi-label data pairs and multi-feature extractors, the proposed loss term integrates with existing task-specific losses by directly constraining gradient directions, thereby enhancing the feature learning process. Extensive theoretical analysis and experimental results demonstrate that, across various domains, datasets, network architectures, and tasks, models incorporating the Lennard-Jones potential loss significantly outperform baseline models without this auxiliary loss in both accuracy and robustness. This approach highlights the potential of physics-inspired loss functions to improve deep learning optimization.

## 1 INTRODUCTION

The second law of thermodynamics (Clausius, 1850) posits that in an isolated system, without external interference, the system will naturally progress toward a state of disorder and equilibrium. Similarly, in deep learning, loss functions act as external forces, guiding randomly initialized, disordered parameters toward a more organized and structured state, thereby enhancing model performance. Task-specific loss functions, like cross-entropy loss (Shannon, 1948), typically target the final-layer features for downstream tasks. While effective at optimizing task performance, these loss functions often overlook the structural refinement of the feature space, which is crucial for enhancing generalization.

Recently, contrastive loss functions (Hadsell et al., 2006) have improved feature space structure by maximizing similarity between similar samples and minimizing similarity between dissimilar ones. Notable examples include the single-modal dual-encoder model in MoCo (He et al., 2020b) and multi-modal learning in CLIP (Radford et al., 2021). However, these methods tend to require substantial additional resources, such as data augmentation, multiple feature extractors, or annotated paired data, which complicate models and increase computational costs (Chen et al., 2020a; Grill et al., 2020a).

In parallel to how physical laws govern the transformation of systems from disorder to order, deep learning optimization aims to evolve chaotic initial features into a low-entropy ordered state. Early research (Rackauckas et al., 2020; Karniadakis et al., 2021; Zhu et al., 2019; Chen et al., 2018; Rackauckas et al., 2020) explored this idea by integrating physical principles into the optimization process. Energy-based (LeCun et al., 2006; Ngiam et al., 2011) and entropy-based (Jaynes, 1957; Grandvalet & Bengio, 2005) methods, inspired by physics, offer alternative perspectives for optimizing feature spaces. Energy-based models cluster similar samples by minimizing energy states, while entropy-based methods increase the entropy of feature distributions to avoid over-fitting. These physics-inspired approaches have shown promise in both self-supervised and unsupervised tasks by

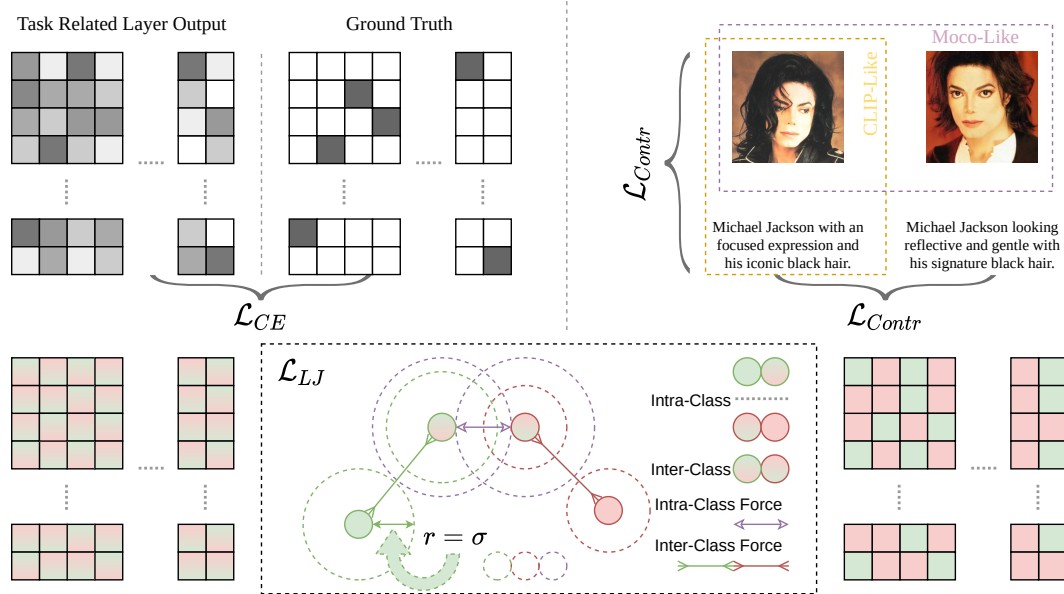

Figure 1: The comparison between SE Loss, contrastive Loss, and our proposed LJ Loss.

improving feature representation, reducing dependence on labeled data, and boosting generalization, thus paving the way for advancements in contrastive learning with simplified model designs.

Building on this foundation, we propose Lennard-Jones (LJ) Loss, an auxiliary loss function aimed at optimizing the feature space. Inspired by the Lennard-Jones potential (Lennard-Jones, 1924) in physics, LJ Loss employs a dual mechanism of "attraction" and "repulsion" within the feature space. When semantic distances between samples are small, LJ Loss applies a repulsive force to avoid over-clustering, while larger distances trigger an attractive force to reduce feature dispersion. As shown in Figure 1, LJ Loss better balances intra-class compactness and inter-class separation compared to traditional Euclidean distance-based methods, without adding inference overhead. Extensive experiments on various computer vision tasks demonstrate substantial performance improvements, especially in cases with small sample sizes and complex intra-class distributions. These findings underscore the potential of LJ Loss in enhancing feature learning and optimization, and suggest promising future research directions. The main contributions of this paper are:

- Propose a gradient direction-constrained loss function based on the Lennard-Jones potential, optimizing feature distributions by regulating sample positions and gradient directions.
- Demonstrate significant performance improvements across various computer vision tasks, including image classification, multi-view learning, and semantic segmentation.
- Analyze feature categories and limitations of LJLoss, providing theoretical insights and proofs to establish a new foundation for optimizing feature representations.

Through these contributions, we demonstrate that a gradient direction-constrained loss based on the Lennard-Jones potential enhances deep learning model performance, promoting the integration of physical models in the development of advanced loss functions.

## 2 RELATED WORK

### 2.1 COMPUTER VISION MODELS

Deep learning models have made significant advancements in computer vision tasks (LeCun et al., 1998; Krizhevsky et al., 2012; Simonyan & Zisserman, 2014; He et al., 2016a; Tan & Le, 2019; Dosovitskiy et al., 2021b; Liu et al., 2021), including image classification, object detection, and semantic segmentation, primarily by learning rich and discriminative feature representations. Classic

architectures like the ResNet (He et al., 2016b) and its variants utilize deep convolutional neural networks (CNNs) to hierarchically extract features from images, aiming to distinguish between different classes (inter-class variation) while minimizing differences within the same class (intra-class variation). These models, however, typically lack mechanisms to handle more subtle intra-class variations — the differences between instances of the same class that are not necessarily captured by maximizing inter-class variance alone. For example, in image and point cloud processing, models can effectively separate different object categories but may struggle with differentiating nuanced intra-class differences, such as variations in pose, illumination, or style among objects of the same category. This limitation suggests the need for more refined approaches that consider both inter-class and intra-class relationships within the feature space.

## 2.2 CONTRASTIVE LEARNING

Contrastive learning has become a powerful approach for self-supervised representation learning (Chen et al., 2020b; He et al., 2020a; Grill et al., 2020b; Khosla et al., 2020), where the objective is to learn meaningful features without direct supervision by maximizing agreement between positive pairs and minimizing agreement between negative pairs. This method relies on explicitly defining what constitutes a "positive" sample (typically, different augmented views of the same instance) and a "negative" sample (different instances altogether), often requiring manual or heuristic-based labeling of these pairs. While contrastive learning excels at learning inter-class distinctions by pulling positive samples closer together and pushing negative samples apart in the feature space, it inherently lacks mechanisms to handle intra-class consistency.

The fundamental limitation of contrastive learning lies in its focus on distinguishing samples based on pre-defined positive and negative pairs. This binary relationship does not inherently consider the nuanced variations that may exist within a single class, such as subtle differences in shape, texture, or style among samples that share the same label. Consequently, contrastive learning methods are not designed to ensure that the features extracted within a single class maintain semantic coherence and meaningful alignment. This gap limits their effectiveness in tasks that require a finer understanding of intra-class structure and variability, especially when dealing with complex or highly variable data distributions.

## 2.3 APPLICATIONS OF THEORETICAL PHYSICS IN DEEP LEARNING

In recent years, theoretical physics has inspired several innovative applications in deep learning, providing new methods for model design and optimization. For example, concepts from statistical mechanics, such as energy minimization, have been used to develop energy-based models that better capture underlying data distributions and improve robustness against adversarial attacks. The Ising model (Ising, 1925), commonly used in physics to describe ferromagnetism, has been adapted to analyze and optimize neural networks, drawing parallels between the dependencies among neurons and the interactions of magnetic spins. Additionally, ideas from quantum mechanics, such as the Feynman path integral (Feynman, 1948), have influenced new approaches to probabilistic reasoning and uncertainty estimation in machine learning. These applications demonstrate that insights from physics can offer fresh perspectives on deep learning, particularly in enhancing model robustness and interpretability. However, most of these applications do not directly address the structuring of feature spaces or the refinement of relationships between individual samples within those spaces, leaving room for new approaches that leverage physical principles in more targeted ways.

## 3 THEORY

This section introduces the theoretical foundation of a loss function inspired by the Lennard-Jones potential (LJ) and its application to deep learning. We start by formalizing the problem of feature optimization and discussing the limitations of traditional contrastive learning methods. Next, we introduce the concepts of global and local features, which are crucial for modeling interactions within the feature space as part of the same physical system. This perspective allows us to apply the Lennard-Jones potential effectively in controlling feature representations (a detailed proof is provided in the appendix). Finally, we provide the physical background of the Lennard-Jones po-

tential, which describes the interactions between microscopic particles, laying the groundwork for our proposed loss function.

## 3.1 FORMALIZATION AND PROBLEM DEFINITION

In the context of feature optimization for deep learning, consider a dataset $\mathcal{D} = \{x_i\}_{i=1}^{N}$, where $x_i \in \mathbb{R}^d$ and $N$ represent the $i$-th sample and the total number of samples, respectively. $d$ is the dimensionality of the samples. A feature extractor $f : \mathbb{R}^d \rightarrow \mathbb{R}^m$ maps each sample $x_i$ to a vector in the feature space, denoted as $\mathbf{f}_i = f(x_i) \in \mathbb{R}^m$, where $m$ is the dimensionality of the feature space.

The similarity between two samples $i$ and $j$ in the feature space can be quantified by the cosine of the angle $\theta_{ij}$ between their feature vectors:

$$\cos \theta_{ij} = \frac{\mathbf{f}_i \cdot \mathbf{f}_j}{\|\mathbf{f}_i\| \|\mathbf{f}_j\|}, \tag{1}$$

where $\mathbf{f}_i \cdot \mathbf{f}_j$ denotes the inner product of the feature vectors, and $\|\mathbf{f}_i\|$ and $\|\mathbf{f}_j\|$ denote the Euclidean norms of the feature vectors. A distance measure between the sample pairs can then be defined as

$$r_{ij} = 1 - \cos \theta_{ij}, \quad r_{ij} \in [0, 2]. \tag{2}$$

This distance metric reflects the relative positioning of the sample pairs: when the feature vectors $\mathbf{f}_i$ and $\mathbf{f}_j$ are identical, $r_{ij} = 0$; when they are in opposite directions, $r_{ij} = 2$. However, in certain applications, it is challenging to determine which sample pairs are positive or negative, making it difficult to directly apply traditional contrastive loss functions. Therefore, a new approach is required to guide the optimization of feature representations from their initial state to an optimal state.

To model the interactions among features effectively, we consider them as entities within the same physical system. This perspective necessitates distinguishing between global and local features, which plays a critical role in how we apply the Lennard-Jones potential to feature optimization (the specific proof is provided in the appendix).

## 3.2 GLOBAL AND LOCAL FEATURES DEFINITION

Global features refer to the comprehensive information extracted from all elements of an input sample, reflecting the overall characteristics of the sample. Unlike local features, global features are not restricted to specific regions or subsets of the input. Instead, they encapsulate high-level information derived from the entire sample, which can be obtained through various means, such as fully connected layers, pooling operations, or attention mechanisms. In a more generalized form, global features can be expressed as:

$$G = f(X) = g(x_1, x_2, \ldots, x_N),$$

where $X$ is the input sample, $x_i$ represents each element or region in the sample, and $g(\cdot)$ is a general operation (such as pooling, attention, or a combination thereof) that aggregates information from the entire input, with no assumption that it must be an average.

Local features, in contrast, focus on a subset of the input sample and are typically extracted by earlier layers of a neural network. Convolution is a common method for generating local features (see Appendix B), as its receptive field generally cannot cover the entire sample. For example, in the case of convolutional layers, the extraction of local features can be described mathematically as:

$$L_{m,n} = \sum_{i=0}^{k-1} \sum_{j=0}^{k-1} w_{i,j} \cdot x_{m+i,n+j},$$

where $w_{i,j}$ denotes the convolutional kernel weights, $x_{m+i,n+j}$ corresponds to the local region of the input sample centered at position $(m, n)$, and $k$ is the kernel size.

However, local features can also be derived through other methods that focus on specific regions or patches of the input, depending on the architecture of the model used. The key difference between global and local features is that local features capture finer details within small regions, while global features integrate information from the entire input.

### 3.3 LENNARD-JONES POTENTIAL

The Lennard-Jones (LJ) potential is a widely used model to describe intermolecular interactions, especially van der Waals forces (van der Waals, 1873). The classical $12 - 6$ form of the potential is given by:

$$V(r) = 4\epsilon \left[ \left( \frac{\sigma}{r} \right)^{12} - \left( \frac{\sigma}{r} \right)^{6} \right],$$ (3)

where $r$ is the distance between particles, $\epsilon$ represents the potential well depth, and $\sigma$ defines the equilibrium distance. This form captures the balance between short-range repulsion, dominated by the $\left( \frac{\sigma}{r} \right)^{12}$ term, and long-range attraction, governed by the $\left( \frac{\sigma}{r} \right)^{6}$ term.

To generalize this potential for more flexible modeling, we can extend the $12 - 6$ form to a $2n - n$ form:

$$V(r) = 4\epsilon \left[ \left( \frac{\sigma}{r} \right)^{2n} - \left( \frac{\sigma}{r} \right)^{n} \right],$$ (4)

where $n$ is a positive integer that controls the steepness of the repulsive and attractive terms. This generalization allows tailoring the interaction profile to specific systems by adjusting $n$.

The force derived from this generalized potential is given by:

$$F(r) = -\frac{dV(r)}{dr} = \frac{4n\epsilon}{r} \left[ 2 \left( \frac{\sigma}{r} \right)^{2n} - \left( \frac{\sigma}{r} \right)^{n} \right].$$ (5)

At $r = \sigma$, the system is in equilibrium with zero net force. For $r < \sigma$, the repulsive term dominates, preventing overlap, while for $r > \sigma$, the attractive term dominates, drawing particles closer. This generalization is useful for capturing a wider range of physical behaviors, from hard-sphere interactions to softer profiles.

In both its classical and generalized forms, the Lennard-Jones potential remains a robust framework for modeling intermolecular forces, balancing short-range repulsion with long-range attraction, and providing flexibility for different physical systems.

## 4 METHODOLOGY OF LJ LOSS FUNCTION

Inspiring by the Lennard-Jones potential, the proposed loss function models molecular interactions, to address limitations in existing contrastive learning approaches. Specifically, the LJ Loss function introduces both inter-class repulsive forces and intra-class attractive forces, promoting well-separated inter-class features while ensuring compact intra-class distributions. Unlike contrastive learning, the LJ Loss function does not rely on predefined positive and negative pairs but instead constructs interaction forces based on the distances between sample representations.

In the feature optimization context, the relative distance between two samples $i$ and $j$ is expressed as $r_{ij} = 1 - \cos\theta_{ij}$, where $\theta_{ij}$ is the angle between feature vectors $\mathbf{f}_i$ and $\mathbf{f}_j$. The generalized Lennard-Jones potential can then be applied as:

$$V_{ij} = 4\epsilon \left( \frac{\sigma^{2n}}{r_{ij}^{2n}} - \frac{\sigma^{n}}{r_{ij}^{n}} \right), \quad \mathcal{L}_{\text{LJ}} = \frac{1}{N^2} \sum_{i=1}^{N} \sum_{j=i+1}^{N} V_{ij},$$ (6)

where $n$ is a positive integer controlling the decay rate of the interaction, $r_{ij} = 1 - \cos\theta_{ij}$ represents the distance between two samples, and $\epsilon$ and $\sigma$ are constants that regulate the strength and range of the potential.

The gradient of $V_{ij}$ with respect to the angle $\theta_{ij}$ is computed as:

$$Grad. = \frac{\partial V_{ij}}{\partial \theta_{ij}} = -4\epsilon \sin\theta_{ij} \left( \frac{2n\sigma^{2n}}{(1-\cos\theta_{ij})^{2n+1}} + \frac{n\sigma^n}{(1-\cos\theta_{ij})^{n+1}} \right). \tag{7}$$

Notably, in Eq.5, the derivative of the Lennard-Jones potential corresponds to the force exerted on the particles. Similarly, the gradient in Eq.7 can be interpreted as a force acting on the feature vectors. Specifically, for small angles $\theta_{ij}$, where features are more similar, attractive forces dominate, drawing intra-class samples closer together. Conversely, at larger angles, repulsive forces prevail, pushing inter-class samples further apart and thereby enhancing class separation.

By minimizing the LJ Loss, the method ensures a feature space where intra-class samples form compact clusters and inter-class samples are well-separated. This physically inspired loss function provides an effective mechanism for optimizing feature representations without requiring predefined positive and negative pairs, offering a flexible solution to representation learning challenges.

For classification tasks, where only intra-class interactions are needed, each vector experiences an attractive force from similar vectors, converging to a stable state similar to physical systems. Repulsive forces prevent excessive overlap when vectors become too close. However, in tasks requiring inter-class interaction, such as segmentation, attractive forces between distinct classes with low similarity are inappropriate, and only repulsive forces should apply. Thus, we have made adaptive modifications to LJ Loss for multi-class tasks. The adaptive equation is as follows:

$$\bar{V}_{ij} = 4\epsilon \left( \frac{\sigma^{2n}}{r_{ij}^{2n}} - \cos\theta_{ij} \times \frac{\sigma^n}{r_{ij}^n} \right), \quad \cos\theta_{ij} \in [-1, 1] \tag{8}$$

where the additional similarity $\cos\theta_{ij}$ denotes as multiplication operator. Specifically, when $f_i$ and $f_j$ represents different classes resulting in a negative number of the $\cos\theta_{ij}$, the effect of LJ Loss is manifested as a repulsive resultant force. Evidence of the necessity of the proposed loss is provided in the Appendix A.

## 5 EXPERIMENT

We evaluate LJ Loss on 2D image recognition and 3D point cloud classification and segmentation tasks. Using visualizations, we highlight its performance across tasks. Various hyperparameter settings (see in Appendix E.2 with implement code), and it's sensitive in Appendix E.4) were tested to ensure a comprehensive evaluation. All experiments ran on up to two RTX 4090 GPUs with a maximum of 24 CPU threads. Classic backbone networks were used, covering a range of architectures. To ensure result consistency, the random seed was fixed at 1, implement details see Appendix E.3. The baseline performance with default loss functions was compared to LJ Loss, focusing on training stability, test set performance, and loss function reduction.

### 5.1 2D IMAGE RECOGNITION

We evaluate Vision Transformer (ViT) (Dosovitskiy et al., 2021a) and ResNet (He et al., 2016b) variants on CIFAR-100 (Krizhevsky & Hinton, 2009) and TinyImageNet (Deng et al., 2009) datasets. CIFAR-100 has 50,000 training and 10,000 test images across 100 classes (32x32 resolution), while TinyImageNet has 100,000 training and 10,000 test images across 200 classes (64x64 resolution). Only random flipping is used for data augmentation during training.

**Direct Classification** The design principle of the backbone network variants is to simulate underfitting, proper fitting, and overfitting scenarios (proof in Appendix D), in order to assess the impact of LJ Loss on model performance under these conditions and evaluate its applicability. The specific parameters of the ViT shown in Table 1 and ResNet variants are detailed in the Appendix C.

Table 1: Comparison of ViT Models with and without LJ Loss in TinyImageNet. Dim refers to the hidden layer dimension. MLP Ratio is 4 for all models. Accuracies are in percentages. $\Delta t$ represents the percentage of additional training time required for the LJ Loss version compared to the baseline. Bold values in parentheses indicate the percentage improvement brought by LJLoss.

| Model | Head | Layer | Patch | Dim | Train Acc | Val Acc | Max Val Acc | $\Delta t$ |
|---|---|---|---|---|---|---|---|---|
| ViT-S/8 | 6 | 6 | 8 | 192 | 91.64(**+0.12**) | 38.50(**+0.78**) | 41.74(**+1.53**) | 9.62% |
| ViT-S/16 | 6 | 6 | 16 | 192 | 93.42(**+2.06**) | 39.93(**+2.89**) | 44.19(**+0.30**) | 7.31% |
| ViT-B/8 | 12 | 12 | 8 | 384 | 90.41(**+2.03**) | 37.69(**+2.50**) | 41.27(**+2.04**) | 11.84% |
| ViT-B/16 | 12 | 12 | 16 | 384 | 92.94(**+2.45**) | 39.53(**+0.92**) | 42.67(**+1.78**) | 3.75% |

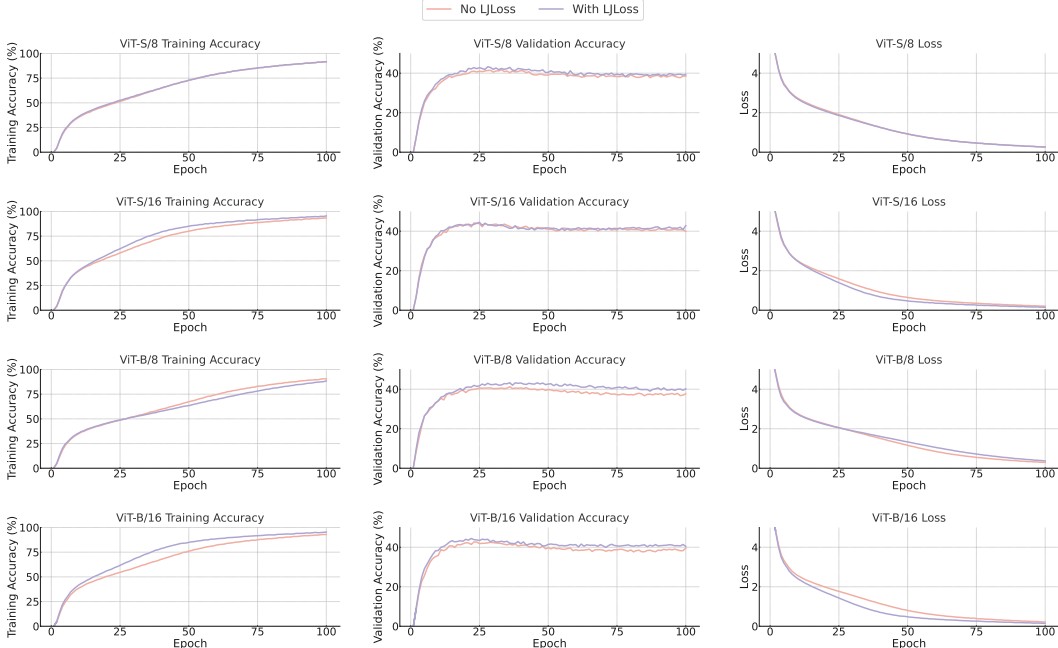

Figure 2: Log visualization of different ViT settings on TinyImageNet.

To address the challenge of LJ Loss struggling to find optimal solutions during fine-tuning—which hampers effective gradient descent in downstream tasks—we trained the ViT model from scratch. Without extra pre-training data, ViT significantly underperforms compared to CNN-based methods that incorporate inductive biases. We recorded test performance and loss at the end of each epoch; Figure 2 visualizes the training of four variants from Table 1 on TinyImageNet.

Models incorporating LJ Loss consistently outperformed the baseline in both training and testing. Larger patches led to a more substantial decrease in loss due to their richer global information and enhanced interactions within attention layers. In smaller ViT-S configurations, we observed slight performance improvements attributed to underfitting from over-parameterization. Even in these underfitting scenarios, models using LJ Loss were less affected and showed greater improvements over the baseline, highlighting LJ Loss's ability to effectively utilize limited information.

**Multi-View Classification** We constructed a MLP model on CIFAR-10 (Krizhevsky & Hinton, 2009). First, the RGB channels were converted to HSV, with each of the three channels carrying distinct but complete RGB information. Each channel was flattened into a 1D vector and processed by three identical fully connected layers, followed by fusion and classification. The fusion methods include channel averaging and concatenation. This approach can be considered a multi-view image classification model, where each of the three feature extraction heads captures global information. Based on this framework, we evaluated the potential of LJLoss in multi-view learning, with results shown in Table 2.

Table 2: Performance of MLP models on CIFAR-10, with accuracy reported as percentages. Bold values indicate improvements from LJLoss over the baseline. The two results under each accuracy column represent average fusion and concat fusion, respectively.

| Layers | Training Accuracy | Test Accuracy |
|---|---|---|
| 0 | 44.19 (+**0.04**) / 45.77 (+**0.11**) | 38.93 (+**0.51**) / 37.71 (+**1.15**) |
| 1 | 74.79 (−**0.12**) / 89.77 (−**0.30**) | 54.11 (+**0.25**) / 53.12 (+**0.67**) |
| 2 | 79.83 (−**0.29**) / 92.92 (+**0.34**) | 53.29 (+**1.12**) / 52.56 (+**1.31**) |
| 3 | 81.91 (+**0.06**) / 93.92 (+**0.20**) | 51.63 (+**1.20**) / 50.71 (+**1.88**) |

The results indicate that the model performs best with a single hidden layer. Throughout training, test accuracy did not decrease, suggesting that the issue is not overfitting but rather over-parameterization. Compared to average fusion, concat fusion fits the training set better but results in lower test performance, likely due to the threefold increase in the final classification layer's parameters, making convergence difficult with limited training data. Across various scenarios, LJLoss directly optimizes the final convergence of multi-view global features. Notably, under over-parameterization, LJLoss significantly improves model performance, mitigating its adverse effects.

Table 3: Performance of ViT models on CIFAR-100. All accuracy values are percentages. Bold values in parentheses indicate the percentage improvement brought by LJLoss.

| Model | Train Acc | Test Acc | Max Test Acc |
|---|---|---|---|
| ViT-S/8 | 97.16 (−**0.46**) | 56.22 (+**0.66**) | 56.78 (+**0.58**) |
| ViT-S/16 | 97.31 (+**0.13**) | 52.68 (+**0.87**) | 54.18 (+**0.74**) |
| ViT-B/8 | 96.38 (+**0.04**) | 54.64 (+**1.34**) | 54.64 (+**1.54**) |
| ViT-B/16 | 96.35 (+**0.55**) | 52.18 (+**1.05**) | 54.00 (+**0.45**) |

We conducted experiments with the ViT model on the CIFAR-100 (Krizhevsky & Hinton, 2009) dataset to further validate LJLoss in mitigating over-parameterization, as shown in Table 3. In some cases, training accuracy decreased while test accuracy improved due to high feature homogeneity, leading to conflicts between LJLoss and the downstream loss. Removing LJLoss increased training accuracy but reduced test accuracy, underscoring the harmful effects of over-parameterization. This strengthens the evidence for LJLoss's performance benefits. Larger models are more affected due to higher feature dimensionality, resulting in lower performance than ViT-S under patch consistency. LJLoss not only improves general performance but also significantly benefits over-parameterization networks, supporting the conclusions in Table 2.

## 5.2 3D POINT CLOUD UNDERSTANDING

We apply LJ Loss to point cloud backbones based on different methods, then test and compare their performance across various downstream tasks. The first task is the ModelNet-40 (Qiu et al., 2022) classification benchmark, which is a point cloud dataset generate from CAD models, consisting of 40 classes, with 9,843 training samples and 2,468 validation samples. The second task is the Scannet v2 (Dai et al., 2017) semantic segmentation benchmark, a point cloud dataset reconstructed from RGB-D frames, comprising 1,201 training scenes and 312 validation scenes.

LJ Loss is incorporated into high-level feature interactions, where each point aggregates semantic information across layers. We conducted classification experiments on ModelNet-40 with Point-Net (Qi et al., 2017a), PointNet++ (Qi et al., 2017b), PointMLP (Ma et al., 2022), and Point Transformer V3 (PTv3) (Wu et al., 2024), showing that LJ Loss improved accuracy with minimal training overhead (Table 4). For segmentation, we adapted LJ Loss to PTv3 on the ScanNet v2 benchmark. After the encoder, points with similar semantics experience both attraction and repulsion forces, while dissimilar points are influenced by repulsion only, reducing cosine distances between similar points. This method yielded superior results, confirming its efficacy. Figure 3 provides a partial visualization of the training process. LJ Loss showed stable performance improvements in both accuracy and loss reduction, achieving faster convergence with comparable training time to the baseline.

Table 4: Performance (percentage for OA or mIoU) and training details for different 3D Backbone on various point cloud datasets. $\Delta t$ represents the percentage of additional training time required for the LJ Loss version compared to the baseline. PointMLP and PTv3 are trained on our settings.

| Backbone | Method | ModelNet40 (OA) | | Scannet v2 (mIoU) | |
|---|---|---|---|---|---|
| | | Performance | $\Delta t$ | Performance | $\Delta t$ |
| PointNet | MLP | 89.2 (**+2.3**) | 5.68 | - | - |
| PointNet++ | MLP | 91.9 (**+1.3**) | 6.12 | - | - |
| PointMLP | MLP | 92.3 (**+0.6**) | 6.00 | - | - |
| PTv3 | Transformer | 92.6 (**+0.4**) | 4.82 | 77.6 (**+0.4**) | 6.57 |

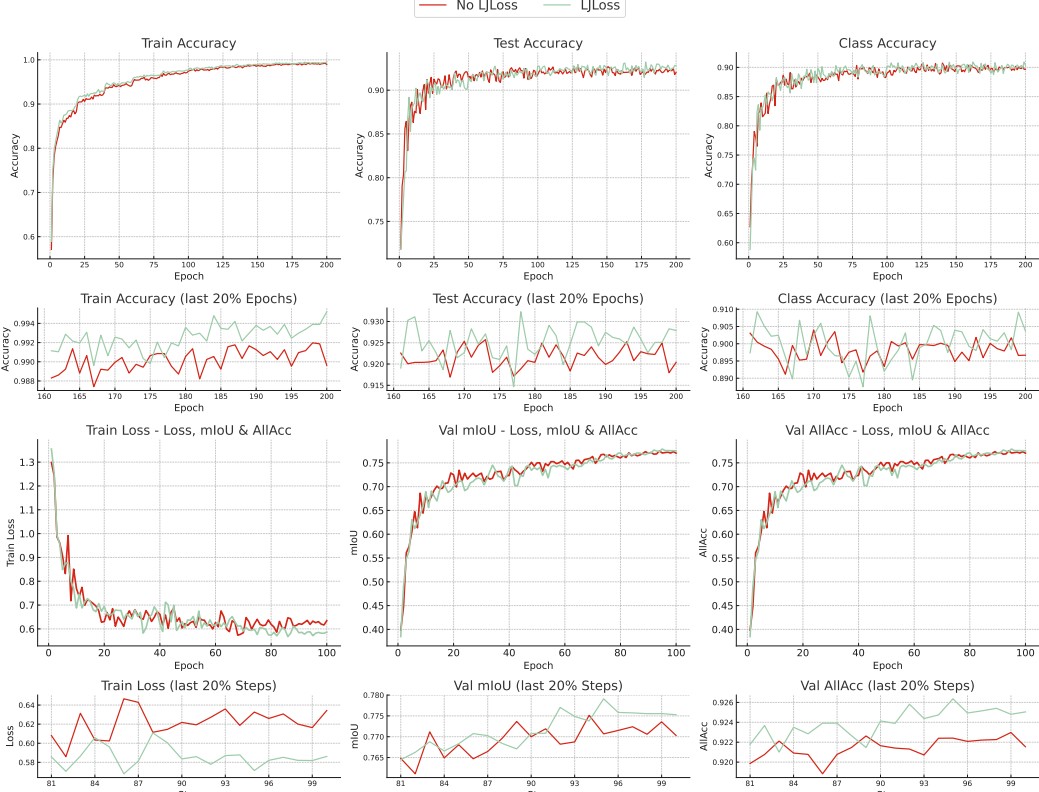

Figure 3: The visualized comparison encompasses two experimental sets. The inset highlights the final 20% of epochs for clearer comparison. The first set reports Train Accuracy, Test Accuracy, and Class Accuracy of PointNet++ on the ModelNet-40 classification task. The second set reports Train Loss, validation mIoU, and validation Accuracy of PTv3 on the Scannet v2 segmentation task.

## 6 CONCLUSION AND LIMITATION

This paper theoretically and experimentally demonstrates the feasibility of applying physical laws to feature re-representation under known initial and final states. The improvements are reflected in comprehensive performance, mitigating the impact of over-parameterization and constraining the model's tendency toward overfitting. Introducing the Lennard-Jones potential as a loss function does not add significant burden to the model and often accelerates convergence, saving time. However, its computational cost grows quadratically with the number of feature channels. Additionally, the globality and parameter combinations, while theoretically sound, may impose overly strict or irrelevant physical constraints. Just as the Lennard-Jones equation is unsuitable for all systems, it cannot handle highly skewed potential wells. Future work will aim to address these limitations.

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

# A   WHY LENNARD-JONES POTENTIAL OVER OTHER PHYSICAL EQUATIONS

We selected the Lennard-Jones (LJ) potential over other physical equations because it effectively guides the feature space optimization toward the intended objective of balancing intra-class compactness and inter-class separation. Other physical equations, such as Coulomb's law (Coulomb, 1785) and gravitational models (Newton, 1687), are not suitable for achieving this goal due to their inherent limitations in describing the relationships between features in a manner conducive to our optimization process. Moreover, these alternatives introduce challenges in quantifying properties like mass or charge, which do not have direct analogs in deep learning feature representations.

For instance, Coulomb's law is expressed as:

$$F = \frac{k_e q_1 q_2}{r^2}$$

where $q_1$ and $q_2$ represent electric charges, and $r$ is the distance between them. Coulomb's force is restricted to binary interactions—either attractive or repulsive, depending on the charge signs. In the context of deep learning, feature representations do not have a natural equivalent to electric charge, making it difficult to apply this equation in a meaningful way. Furthermore, Coulomb's law, with its inverse-square relationship, can result in extreme gradient magnitudes when applied to optimization, leading to potential instability, such as gradient explosion or vanishing.

Similarly, the gravitational force equation:

$$F = \frac{G m_1 m_2}{r^2}$$

where $m_1$ and $m_2$ denote masses, suffers from similar limitations. The gravitational force only describes attraction between masses, lacking any repulsive counterpart. This makes it unsuitable for controlling both intra-class compactness and inter-class separation, as required in feature space optimization. Additionally, mass, like charge, is not a well-defined property in deep learning models, complicating its application. The lack of repulsion in the gravitational model would result in features clustering too tightly, which is undesirable for maintaining separation between distinct classes.

In contrast, the Lennard-Jones potential (Lennard-Jones, 1924) is expressed as:

$$V(r) = 4\epsilon \left[ \left( \frac{\sigma}{r} \right)^{12} - \left( \frac{\sigma}{r} \right)^6 \right]$$

This potential provides a dual mechanism of attraction and repulsion: the $\left( \frac{\sigma}{r} \right)^{12}$ term introduces a strong repulsive force at short distances, preventing over-clustering of features, while the $\left( \frac{\sigma}{r} \right)^6$ term introduces an attractive force at larger distances, preventing excessive feature dispersion. This balance of forces directly aligns with the goals of feature space optimization in deep learning, where we seek to maintain class separation while ensuring compact intra-class representations. Additionally, the LJ potential operates smoothly without the extreme gradients associated with inverse-square laws, resulting in more stable training dynamics.

Moreover, the Lennard-Jones potential avoids the need for defining abstract physical properties like charge or mass. It solely relies on distances between feature representations, making it directly applicable to our task without requiring arbitrary mappings to physical quantities that do not naturally fit the problem context. The mathematical simplicity and computational efficiency of the LJ potential further ensure that it does not introduce significant overhead, making it an ideal candidate for our deep learning model optimization.

In summary, alternative physical equations such as Coulomb's law and gravitational models are unsuitable for achieving our optimization goals due to their single-faceted nature (either attraction or repulsion), difficulties in quantifying properties such as charge or mass, and their tendency to produce unstable gradients. The Lennard-Jones potential, with its balanced dual mechanism of attraction and repulsion, low computational complexity, and compatibility with feature space dynamics,

Figure 4: The receptive field size of different CNN layers. Each box represents the effective receptive field.

provides a superior solution for the task at hand. Therefore, additional ablation studies on other physical equations are unnecessary.

## B    LOCALITY REASONING AND DEFINITION IN CONVOLUTIONAL NEURAL NETWORKS

Convolutional Neural Networks (CNNs) extract features through localized convolutional operations, where each layer's receptive field expands progressively. This expansion, however, remains constrained by the size of the convolutional kernel and the stride. The size of the receptive field $R_L$ at layer $L$ can be computed recursively using the following formula:

$$R_L = R_{L-1} + (k_L - 1) \cdot \prod_{i=1}^{L-1} s_i,$$

where: - $R_L$ is the receptive field of layer $L$, - $R_{L-1}$ is the receptive field of the previous layer, - $k_L$ is the kernel size at layer $L$, - $s_i$ is the stride at layer $i$.

For an input image of size $H_0 \times W_0$, the spatial size $H_L \times W_L$ at layer $L$ can be calculated as:

$$H_L = \left\lfloor \frac{H_{L-1} - k_L + 2p_L}{s_L} \right\rfloor + 1,$$

$$W_L = \left\lfloor \frac{W_{L-1} - k_L + 2p_L}{s_L} \right\rfloor + 1,$$

where $p_L$ is the padding at layer $L$.

Taking ResNet as an example, as shown in Figure 4, the first layer utilizes a convolutional kernel of size $7 \times 7$ with a stride of 2 and padding of 3. The effective receptive field at this layer is:

$$R_1 = 7.$$

Subsequent convolutional layers employ $3 \times 3$ kernels with a stride of 2. Assuming there are $L$ convolutional layers, we can iteratively compute the receptive field. For simplicity, consider that each layer has the same kernel size and stride:

$$R_L = R_{L-1} + (k-1) \cdot \prod_{i=1}^{L-1} s_i.$$

After multiple layers, the receptive field may still be significantly smaller than the input image size, indicating that each neuron in the deeper layers only "sees" a localized region of the input.

Global Average Pooling (GAP) is applied at the end of the convolutional layers to produce a global feature vector. The GAP operation can be expressed as:

$$G = \frac{1}{H_L \times W_L} \sum_{i=1}^{H_L} \sum_{j=1}^{W_L} F_{i,j},$$

where $F_{i,j}$ represents the feature map at position $(i, j)$ in the last convolutional layer. This operation aggregates spatial information to form a global representation.

## C  PHYSICAL SPACE ANALOGY AND LIMITATIONS OF LOCAL FEATURES

In theoretical physics, the state of a particle system is described by a wave function $\Psi(\mathbf{r}, t)$, which contains all the information about the system. The probability density of finding a particle at position $\mathbf{r}$ and time $t$ is given by:

$$P(\mathbf{r}, t) = |\Psi(\mathbf{r}, t)|^2$$

The evolution of the wave function is governed by the Schrödinger equation (Schrödinger, 1926):

$$i\hbar \frac{\partial \Psi(\mathbf{r}, t)}{\partial t} = \hat{H} \Psi(\mathbf{r}, t)$$

where $\hbar$ is the reduced Planck constant and $\hat{H}$ is the Hamiltonian operator representing the total energy of the system.

In a multi-particle system, the total wave function is a combination of individual particle wave functions. For non-interacting particles, the total wave function is the product of individual wave functions:

$$\Psi_{\text{total}}(\mathbf{r}_1, \mathbf{r}_2, ..., \mathbf{r}_n, t) = \prod_{i=1}^{n} \Psi_i(\mathbf{r}_i, t)$$

When particles belong to different systems or are isolated (non-interacting), their combined wave function does not yield meaningful interactions. This principle mirrors the behavior of local features in CNNs.

In CNNs, each local feature can be analogized to a particle's wave function $\psi_i(\mathbf{r})$, confined to a specific receptive field. Due to the locality of the receptive field, these features are isolated, similar to particles in separate non-interacting systems. The overall feature representation can be considered as:

$$\Psi_{\text{CNN}} = \{\psi_1(\mathbf{r}), \psi_2(\mathbf{r}), ..., \psi_n(\mathbf{r})\}$$

where each $\psi_i(\mathbf{r})$ is localized and does not inherently interact with other local features.

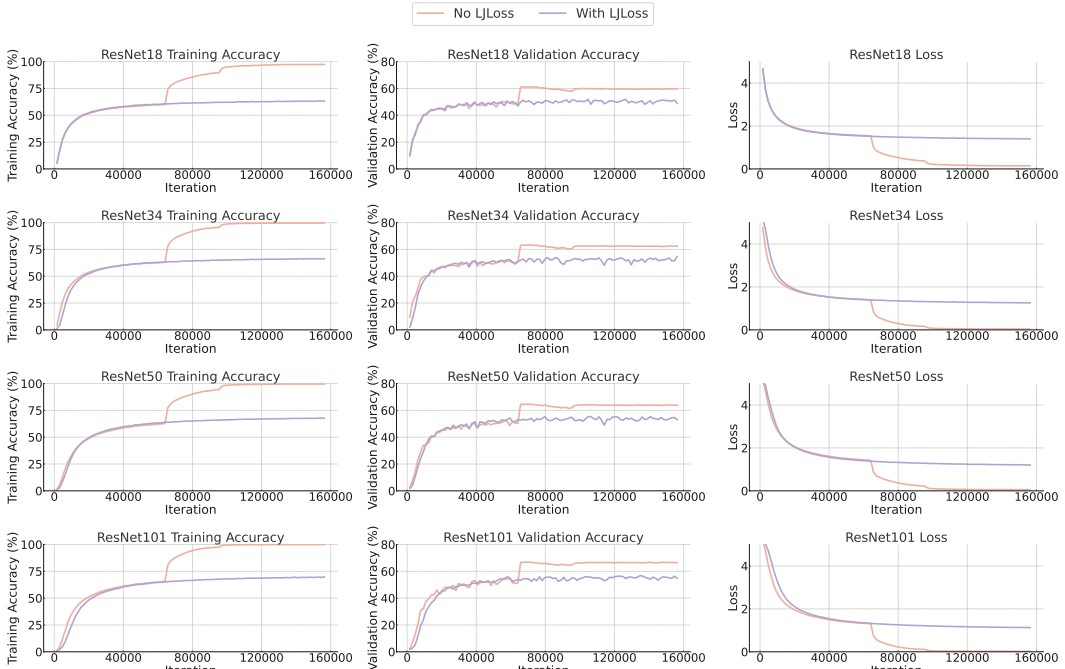

Figure 5: Performance of ResNet on TinyImageNet.

The rearrangement or combination of these local features without proper integration mechanisms does not produce meaningful global representations. This is analogous to attempting to create interactions between particles in separate quantum systems without any coupling; the total system remains a simple aggregation of independent states.

Moreover, the Lennard-Jones potential, commonly used to describe interactions between particles at the molecular level, is given by:

$$V(r) = 4\epsilon \left[ \left( \frac{\sigma}{r} \right)^{12} - \left( \frac{\sigma}{r} \right)^{6} \right]$$

where: - $r$ is the distance between two particles, - $\epsilon$ is the depth of the potential well, - $\sigma$ is the finite distance at which the inter-particle potential is zero.

In the context of feature interactions, applying the Lennard-Jones potential assumes that all feature elements are within the same interactive space, and their distances $r$ are meaningful and comparable. However, due to the localized receptive fields in CNNs, the "distance" between local features in different receptive fields lacks meaningful interpretation, as these features are not within the same interactive "space" or context.

Figure 5 shows the performance of ResNet on the TinyImageNet dataset. Since the image resolution of TinyImageNet is only 64, we adjusted the kernel size of the first convolutional layer from 7 to 3, with a corresponding stride reduction from 3 to 2. Before the first learning rate adjustment (see Appendix for adjustment strategy), the impact of LJ Loss on the model was minimal, with only a slight performance disadvantage observed. After the learning rate adjustment, the group with LJ Loss failed to learn more precise and detailed feature representations. This further validates the limitations of LJ Loss, which is based on global feature optimization, in tasks involving local receptive fields in CNNs. The global potential energy calculation used by LJ Loss does not effectively optimize the local features extracted by CNNs, as it modifies the distances between different receptive fields, which lack practical physical significance. Consequently, LJ Loss not only fails to enhance performance in the local feature learning of CNNs but also leads to a degradation in the model's feature representation ability after learning rate adjustment, resulting in a decline in overall accuracy.

To model interactions effectively, features must reside within a shared space where their relationships are defined and meaningful. In CNNs, this shared space is not naturally established between local features due to the inherent design of localized receptive fields. Thus, applying potentials like the Lennard-Jones potential without considering the spatial and contextual isolation of local features may not yield beneficial results in feature learning.

Furthermore, integrating local features into a meaningful global representation requires mechanisms that bridge the receptive fields, such as attention mechanisms or feature pyramids, which allow information to flow across different spatial locations. Without such mechanisms, the optimization of local features based on global criteria (e.g., using LJ Loss) may not be effective.

In conclusion, the analogy to physical systems highlights the limitations of treating local features as freely interacting elements within a shared space. The isolation inherent in CNN architectures necessitates careful consideration when designing loss functions or interaction models intended to operate on global feature relationships.

# D OVERFITTING, REDUNDANT FEATURES, AND ANALYSIS OF LJ LOSS

## D.1 OVERFITTING: EXPLANATION AND DEFINITION

Overfitting occurs when a machine learning model becomes overly complex, fitting noise and accidental patterns in the training data, which results in poor generalization performance. To understand the essence of overfitting, it is necessary to clarify the concept of hypothesis space.

The hypothesis space $H$ is the set of all functions the model can choose from. Given a dataset $D = \{(x_i, y_i)\}_{i=1}^N$, which is drawn from an unknown true distribution $P(x, y)$, the goal of machine learning is to find a hypothesis $h \in H$ that minimizes the loss over the true distribution:

$$\mathcal{L}(h) = \mathbb{E}_{(x,y) \sim P}[\mathcal{L}(h(x), y)]$$

In practice, we only have access to a limited training dataset $D_{\text{train}}$, and thus the loss minimization is performed over this finite dataset:

$$\mathcal{L}_{\text{train}}(h) = \frac{1}{N} \sum_{i=1}^{N} \mathcal{L}(h(x_i), y_i)$$

When the hypothesis space $H$ is too large or complex, the model may select a hypothesis $h_{\text{overfit}}$ that fits the training data perfectly but generalizes poorly. In this case, the training loss approaches zero:

$$\mathcal{L}_{\text{train}}(h_{\text{overfit}}) \to 0$$

However, the test loss remains large:

$$\mathcal{L}_{\text{test}}(h_{\text{overfit}}) \gg \mathcal{L}_{\text{train}}(h_{\text{overfit}})$$

The concept of overfitting can be further understood through the Vapnik-Chervonenkis (VC) dimension (Vapnik & Chervonenkis, 1971), which measures the capacity of a model by quantifying its ability to shatter data points in any distribution. A higher VC dimension implies that the model has more capacity to fit arbitrary patterns in the training data. When the model's capacity, represented by the VC dimension, exceeds the inherent complexity of the data distribution, the model tends to overfit. Mathematically, the loss can be decomposed using the bias-variance trade-off:

$$\mathbb{E}[\mathcal{L}] = \text{Bias}^2 + \text{Var} + \sigma^2$$

where Bias refers to the model's error due to simplifying assumptions, Var is the model's sensitivity to fluctuations in the training set, and $\sigma^2$ represents the irreducible noise. In an overfitted model, the variance term dominates, leading to poor generalization performance.

Overfitting also depends on the size of the dataset. On smaller datasets, a model may lack sufficient data to capture the true distribution, leading to overfitting as the model fits the noise in the limited samples. On the other hand, larger models on the same dataset can overfit because they have the capacity to fit overly complex patterns (prior hypotheses) that may not exist in the data distribution.

## D.2  REDUNDANT FEATURES: PHYSICAL ANALOGY AND EXPLANATION

Redundant features arise during overfitting, representing features that do not carry meaningful semantic information. These features can be compared to disordered particles in a physical system, which increase the system's entropy and reduce its overall order. To analyze this phenomenon, we must delve into the concept of entropy, drawing a parallel between physical entropy and the uncertainty in feature spaces.

Entropy in physics is used to measure the disorder in a system and is defined as:

$$S = -k_B \sum_i p_i \log p_i$$

where $p_i$ is the probability of the system being in state $i$, and $k_B$ is the Boltzmann constant. The increase in entropy reflects an increase in the disorder of the system.

This concept can be mapped to the feature space of a neural network. Effective features correspond to ordered particles, representing meaningful information in the model, while redundant features correspond to disordered particles, which increase the system's disorder. When redundant features dominate the model, the total entropy of the feature space increases, leading to greater disorder among the features and reduced generalization performance.

**Transition from Physical Entropy to Information Entropy**

The notion of physical entropy aligns with information entropy, denoted as $H(X)$, which measures uncertainty. It is defined as:

$$H(X) = -\sum_i p(x_i) \log p(x_i)$$

where $p(x_i)$ is the probability of the random variable $X$ taking value $x_i$. In the feature space of a neural network, effective features and redundant features contribute differently to information entropy.

*Effective Feature Entropy $H_{effective}$*: Effective features carry meaningful information, leading to more deterministic predictions. As a result, their entropy is lower, indicating that the model can leverage these features with relative certainty:

$$H_{\text{effective}} = -\sum_{i \in \text{effective}} p(x_i) \log p(x_i)$$

*Redundant Feature Entropy $H_{redundant}$*: Redundant features, on the other hand, typically represent noise or uninformative signals, increasing the model's prediction uncertainty. As redundant features increase, so does their entropy, indicating greater disorder:

$$H_{\text{redundant}} = -\sum_{i \in \text{redundant}} p(x_i) \log p(x_i)$$

In an overfitted model, the majority of the total system entropy $H_{\text{total}}$ comes from the redundant features:

$$H_{\text{total}} = H_{\text{effective}} + H_{\text{redundant}}$$

Effective feature entropy remains relatively stable, while redundant feature entropy increases significantly during overfitting, leading to an overall increase in disorder and poorer generalization.

### D.3 THE PHYSICAL INTERPRETATION OF LJ LOSS AND ITS INAPPLICABILITY IN DEEP LEARNING

In Lennard-Jones potential (LJ Loss), the equation describing the interaction between particles is:

$$V(r) = 4\epsilon \left[ \left( \frac{\sigma}{r} \right)^{12} - \left( \frac{\sigma}{r} \right)^{6} \right]$$

where $r = \|\mathbf{f}_i - \mathbf{f}_j\|$ represents the distance between feature vectors $\mathbf{f}_i$ and $\mathbf{f}_j$. The aim of LJ Loss in deep learning is to adjust the distances between feature vectors, optimizing the overall feature space distribution. The mathematical formulation of LJ Loss is:

$$\mathcal{L}_{\mathrm{LJ}} = \sum_{i \neq j} \left( \frac{A}{\|\mathbf{f}_i - \mathbf{f}_j\|^{12}} - \frac{B}{\|\mathbf{f}_i - \mathbf{f}_j\|^{6}} \right)$$

where $A$ and $B$ are constants that control repulsive and attractive forces between the feature vectors $\mathbf{f}_i$ and $\mathbf{f}_j$.

While LJ Loss helps to structure the feature space by controlling the distances between effective features, it becomes ineffective when applied to redundant features. Redundant features, lacking meaningful semantic information, do not contribute to improving the model's generalization performance. In fact, adjusting the distances between redundant features using LJ Loss can increase the disorder of the system, akin to applying forces between disordered particles in a physical system with high entropy.

Thus, when the feature space is dominated by redundant features due to overfitting, LJ Loss may reinforce these disordered interactions, further worsening the model's performance. This explains why applying LJ Loss in an overfitted model may lead to a further decrease in generalization capability.

### D.4 GRADIENT ANALYSIS AND EXTENDED GRADIENT ANALYSIS

The gradient of LJ Loss with respect to a feature vector $\mathbf{f}_i$ can be computed by taking the partial derivative of the loss function:

$$\nabla_{\mathbf{f}_i} \mathcal{L}_{\mathrm{LJ}} = \sum_{j \neq i} \left( -12 \frac{A(\mathbf{f}_i - \mathbf{f}_j)}{\|\mathbf{f}_i - \mathbf{f}_j\|^{14}} + 6 \frac{B(\mathbf{f}_i - \mathbf{f}_j)}{\|\mathbf{f}_i - \mathbf{f}_j\|^{8}} \right)$$

This gradient directs the movement of feature vectors $\mathbf{f}_i$ relative to other feature vectors $\mathbf{f}_j$. For effective features, the gradient helps structure the feature space by guiding the vectors to meaningful positions. However, for redundant features, the gradient does not lead to meaningful improvements, potentially widening the gap between disordered features and increasing the system's entropy.

**Extended Gradient Analysis**

To analyze how LJ Loss affects the overall optimization process, we examine the total gradient of the combined loss function, including task loss $\mathcal{L}_{\mathrm{task}}$ and LJ Loss $\mathcal{L}_{\mathrm{LJ}}$. The total loss is:

$$\mathcal{L} = \mathcal{L}_{\mathrm{task}} + \lambda \mathcal{L}_{\mathrm{LJ}}$$

where $\lambda$ is a regularization parameter that controls the influence of LJ Loss. The total gradient is given by:

$$\nabla_\theta \mathcal{L} = \nabla_\theta \mathcal{L}_{\mathrm{task}} + \lambda \nabla_\theta \mathcal{L}_{\mathrm{LJ}}$$

The gradient of $\mathcal{L}_{\mathrm{LJ}}$ can be further decomposed into gradients from effective and redundant features:

$$\nabla_\theta \mathcal{L}_{\mathrm{LJ}} = \nabla_\theta \mathcal{L}_{\mathrm{LJ}_{\mathrm{effective}}} + \nabla_\theta \mathcal{L}_{\mathrm{LJ}_{\mathrm{redundant}}}$$

For redundant features, the gradient can be expanded as follows:

$$\nabla_\theta \mathcal{L}_{\text{LJ}_{\text{redundant}}} = \sum_{i \in \text{redundant}} \sum_{j \neq i} \left( -12 \frac{A(\mathbf{f}_i - \mathbf{f}_j)}{\|\mathbf{f}_i - \mathbf{f}_j\|^{14}} + 6 \frac{B(\mathbf{f}_i - \mathbf{f}_j)}{\|\mathbf{f}_i - \mathbf{f}_j\|^8} \right)$$

This gradient shows that when redundant features dominate, the gradient updates are mainly influenced by the disordered interactions between these features. As the number of redundant features increases, the impact of $\nabla_\theta \mathcal{L}_{\text{LJ}_{\text{redundant}}}$ grows, leading to inefficient optimization that wastes resources adjusting the irrelevant features. The total gradient, decomposed into effective and redundant contributions, is:

$$\nabla_\theta \mathcal{L} = \nabla_\theta \mathcal{L}_{\text{effective}} + \nabla_\theta \mathcal{L}_{\text{redundant}}$$

In an overfitted model, the gradient from redundant features dominates, resulting in poor performance. This extended analysis highlights how the improper application of LJ Loss in the presence of redundant features contributes to gradient instability and performance degradation.

### D.5 Chemical Reaction Analogy for Redundant Features and LJ Loss Ineffectiveness

We consider the hydrolysis of ethyl acetate ($CH_3COOCH_2CH_3$) in aqueous solution as a real-world analogy. This reaction typically produces two main products: acetic acid ($CH_3COOH$) and ethanol ($CH_3CH_2OH$). However, due to reaction kinetics and other factors, unwanted by-products can also form, representing redundant features in this analogy.

The main reaction is as follows:

$$CH_3COOCH_2CH_3 + H_2O \rightarrow CH_3COOH + CH_3CH_2OH$$

**Main products**: - Acetic acid ($CH_3COOH$) - Ethanol ($CH_3CH_2OH$)

These main products interact meaningfully in the system, much like effective features in a neural network that improve overall performance through well-structured interactions.

However, alongside these main products, the reaction can generate **by-products** due to incomplete dissociation of ethyl acetate or unwanted side reactions. For example, partial decomposition of ethyl acetate might result in by-products such as acetaldehyde ($CH_3CHO$) or other organic fragments:

$$CH_3COOCH_2CH_3 \rightarrow CH_3CHO + CH_3OH$$

These by-products, analogous to redundant features in deep learning, do not contribute constructively to the reaction and may even interfere with the formation of the desired products. The interaction between the main products is orderly, much like how effective features interact to enhance neural network performance, whereas the interaction between the main products and by-products is random and chaotic, similar to the disordered nature of redundant features in a model.

To draw an analogy to deep learning, we can use the Lennard-Jones potential to model the interactions between the main products and between by-products. For the interaction between acetic acid and ethanol (effective features), the Lennard-Jones potential describes the attraction and repulsion forces that guide the molecules to an optimal distance, forming a stable and meaningful interaction:

$$V_{\text{effective}}(r) = 4\epsilon \left[ \left( \frac{\sigma}{r} \right)^{12} - \left( \frac{\sigma}{r} \right)^6 \right]$$

Where: - $r$ is the distance between acetic acid and ethanol molecules. - $\epsilon$ represents the depth of the potential well, corresponding to the strength of the interaction. - $\sigma$ is the equilibrium distance where the potential is minimized.

On the other hand, by-products like acetaldehyde and methanol (redundant features) do not contribute to a stable interaction with the main products. The Lennard-Jones potential applied to the interaction between by-products and main products would show that these interactions are weak, unstable, or even detrimental to the system's overall stability:

$$V_{\text{redundant}}(r) = 4\epsilon' \left[ \left( \frac{\sigma'}{r} \right)^{12} - \left( \frac{\sigma'}{r} \right)^{6} \right]$$

Where: - $\epsilon'$ and $\sigma'$ are different from those of the main product interaction, indicating a weaker and less significant interaction. - This weak interaction leads to increased disorder in the system, similar to how redundant features fail to contribute meaningful information in a neural network.

Thus, while the interaction between the main products (effective features) is stable and beneficial, the interaction between by-products and main products (redundant features) is ineffective and may disrupt the system's order. In the context of deep learning, applying LJ Loss to redundant features, much like attempting to optimize by-products, does not improve the system's overall performance but rather increases disorder.

# E  IMPLEMENTATION DETAILS

## E.1  SETTING A FIXED RANDOM SEED

To ensure the reproducibility of our experimental results, we fixed the random seed across all libraries and frameworks used in our implementation. Reproducibility is crucial in scientific research to validate findings and enable others to replicate experiments under the same conditions. The following function was employed to set the random seed:

Listing 1: Setting a Fixed Random Seed

```python
def set_seed(seed):
    random.seed(seed)
    np.random.seed(seed)
    torch.manual_seed(seed)          # CPU
    torch.cuda.manual_seed(seed)     # GPU
    torch.cuda.manual_seed_all(seed) # All GPUs
    os.environ['PYTHONHASHSEED'] = str(seed)  # Disable hash
        randomization
    torch.backends.cudnn.deterministic = True    # Ensure
        deterministic convolution algorithms
    torch.backends.cudnn.benchmark = False       # Disable
        benchmarking for reproducibility
```

This function ensures deterministic behavior by setting seeds for Python, NumPy, and PyTorch random number generators, and configuring PyTorch's cuDNN backend for reproducibility.

## E.2  LENNARD-JONES LOSS FUNCTION

We employed a Lennard-Jones (LJ) loss function to encourage diversity in the learned feature representations by penalizing feature vectors that are either too close or too far from each other in the cosine space. The LJ potential is a mathematical model that describes the interaction between a pair of particles and is defined as:

$$V(r) = \left( \frac{\sigma}{r} \right)^{2n} - \left( \frac{\sigma}{r} \right)^{n} \tag{9}$$

where $V(r)$ is the potential energy, $r$ is the distance between particles, $\sigma$ is the distance at which the potential energy is zero (the ideal distance), and $n$ is a parameter that determines the steepness of the potential.

In our implementation, the LJ potential is applied to the cosine distances between feature vectors. The loss function comprises two components: the 'lj_loss' function and the 'LJ Loss' class.

**lj_loss Function**   The 'lj_loss' function computes the Lennard-Jones loss based on normalized feature vectors and their cosine distances.

Listing 2: Lennard-Jones Loss Function

```python
def lj_loss(features, sigma=1.0, n=6, clamp_max=5.0):
    """
    Lennard-Jones loss function using cosine distance for a batch
        of feature vectors.

    Args:
        features: Tensor of shape (b, l, d), where:
                    b: batch size
                    l: sequence length
                    d: feature dimension
        sigma: The distance at which the potential is zero (ideal
            distance in cosine space).
        n: generalized Lennard-Jones potential, e.g., 6
        clamp_max: Clamp the maximum value of loss.

    Returns:
        lj_loss: The computed Lennard-Jones loss for the batch.
    """
    features_normalized = F.normalize(features, dim=2)
    cosine_sim = torch.matmul(features_normalized,
        features_normalized.transpose(1, 2))
    cosine_dist = 1 - cosine_sim
    diag_indices = torch.arange(cosine_dist.size(1),
        device=cosine_dist.device)
    cosine_dist[:, diag_indices, diag_indices] = sigma
    cosine_dist = torch.clamp(cosine_dist, min=1e-3)
    term1 = (sigma / cosine_dist) ** (2 * n)
    term2 = ((sigma / cosine_dist) ** n)
    lj_potential = (term1 - term2)
    lj_potential = torch.clamp(lj_potential, max=clamp_max)
    lj_loss = torch.mean(lj_potential)
    return lj_loss
```

This function calculates the Lennard-Jones loss by normalizing feature vectors, computing cosine similarities and distances, and applying the Lennard-Jones potential with clamping to avoid extreme values.

**LJ Loss Class**   The 'LJ Loss' class integrates the 'lj_loss' function into a PyTorch 'nn.Module', allowing it to be seamlessly incorporated into the training pipeline.

Listing 3: LJ Loss Class

```python
class LJ Loss(nn.Module):
    """
    Default settings of LJ_loss
    """
    def __init__(self, epsilon=0.1, sigma=1.0, n=6,
        clamp_max=5.0):
        super(LJ Loss, self).__init__()
        self.sigma = sigma
        self.n = n
```

```
        self.clamp_max = clamp_max
        self.epsilon = epsilon

    def forward(self, feat):
        return self.epsilon * lj_loss(
            feat,
            sigma=self.sigma,
            n=self.n,
            clamp_max=self.clamp_max
        )
```

The 'LJ Loss' class scales the computed LJ Loss by a factor of 'epsilon', allowing for flexible integration with other loss components during model training.

### E.3 TRAINING PARAMETERS FOR RESNET AND VISION TRANSFORMER

We trained both ResNet and Vision Transformer (ViT) models using specific training parameters to optimize performance on the dataset.

**ResNet Training Parameters**   For the ResNet model, we employed stochastic gradient descent (SGD) (Bottou, 2010) with momentum. In this setup:

- The initial learning rate was set to $lr = 0.1$.
- Momentum of 0.9 was used to accelerate gradient vectors in the right direction.
- A weight decay of $1 \times 10^{-4}$ was applied to prevent overfitting.
- The learning rate scheduler reduced the learning rate by a factor of 0.1 at epochs 41, 61, and 81.

**Vision Transformer Training Parameters**   For the ViT model, we utilized the Adam optimizer (Kingma & Ba, 2014) with weight decay and implemented a learning rate scheduler that combines warm-up and cosine annealing. Details of the configuration:

- The initial learning rate was set to $lr = 1 \times 10^{-3}$.
- The optimizer used default $\beta$ parameters $(0.9, 0.999)$.
- A weight decay of $5 \times 10^{-5}$ was applied.
- A cosine annealing learning rate scheduler with warm-up was used:
  - The learning rate was gradually increased during the first 5 epochs (warm-up period).
  - After the warm-up, the learning rate followed a cosine annealing schedule with a minimum learning rate of $1 \times 10^{-5}$ and a period of 200 epochs.

The combination of warm-up and cosine annealing helps in stabilizing the training in the initial phase and allows for effective learning rate decay over time.

Both models were trained with a batch size of 64 and used cross-entropy loss as the objective function.

### E.4 HYPERPARAMETERS OF LJ LOSS

Due to the high semantic density of images and the sparse representation of point clouds, logically, the potential well of point cloud features is generally larger, requiring a stronger discriminative capacity to distinguish between features. In summary, all the LJ Loss hyperparameters used in the experiments are divided into two groups: for image recognition tasks, the potential well $\sigma$ is set to 0.5 and the exponent $n$ to 6; for point cloud understanding tasks, the potential well $\sigma$ is set to 1.0 and the exponent $n$ to 6.

We replaced the hyperparameters from the image understanding experiments with those used in point cloud understanding experiments, and the results are shown in the figure.

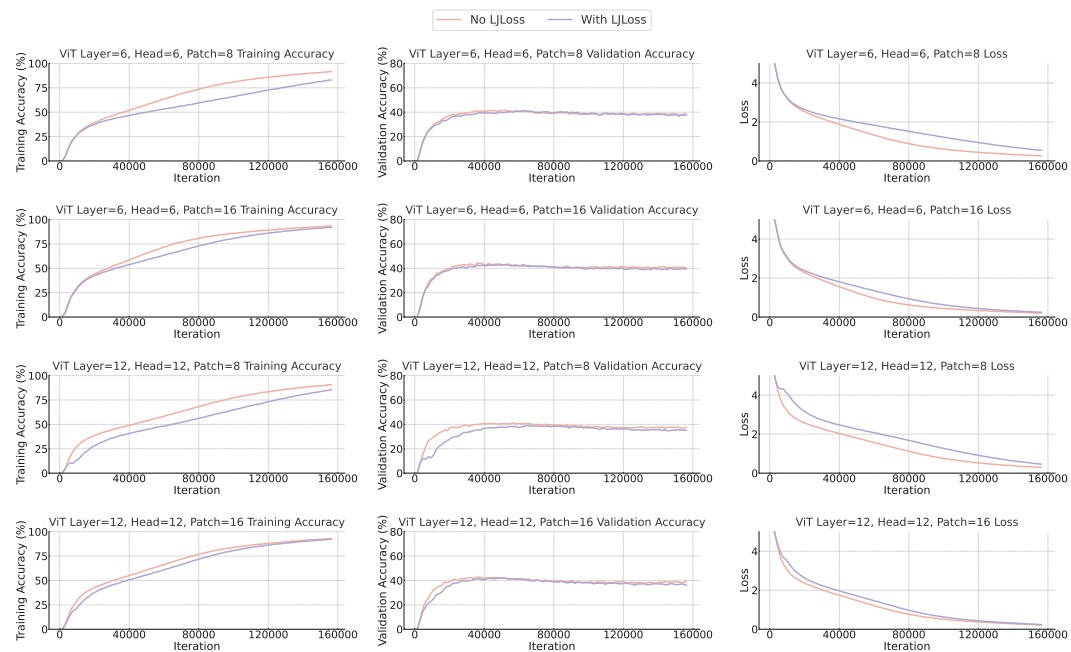

Figure 6: Different settings of VIT in TinyImageNet dataset.

Overall, as shown in Figure 6, setting the potential well too high and the exponent too large imposes extremely high demands on feature discrimination. However, overemphasizing feature distinction may lead to adverse effects, such as model overfitting or reduced generalization capability. It is worth noting that, even under such overemphasis, the relationship between LJ Loss and patch size still holds. Larger patches typically result in a smoother loss function, thereby improving the effectiveness of LJ Loss.

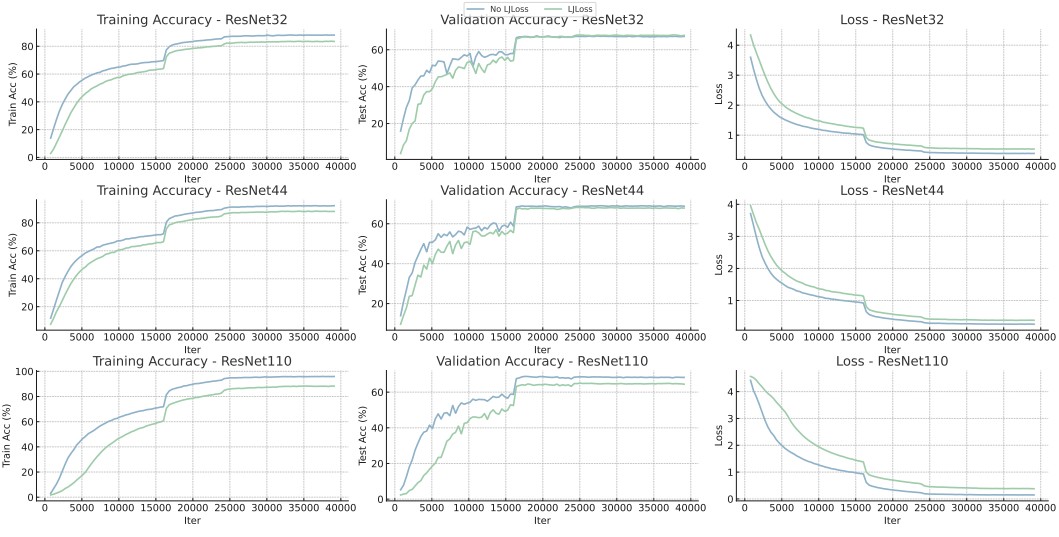

Figure 7: Performance of ResNet in CIFAR-100 dataset.

As the model size increases, the impact of this overemphasis on feature similarity and dissimilarity becomes more pronounced, particularly in ResNet, where the features do not originate from a homogeneous system. We validated the performance of ResNet on the CIFAR-100 dataset to support this observation. As shown in Figure 7, with deeper network layers, the performance difference introduced by LJ Loss increasingly diverges from the baseline.

