# OpenReview forum: "Self-Supervised Feature Re-Representation via Lennard-Jones Potential Loss"
_ICLR.cc/2025/Conference — ICLR 2025 Conference Withdrawn Submission_

### Official Review · Reviewer_qF5S · 2024-10-29

**Soundness:** 2
**Presentation:** 1
**Contribution:** 2
**Rating:** 3
**Confidence:** 4

**Summary:**

This paper proposes a self-supervised feature re-representation technique using the Lennard-Jones potential inspired by molecular interactions as a loss function, aiming to balance intra-class compactness and inter-class separation in feature space. The method enhances feature clustering without predefined positive-negative pairs. Empirical results on various vision tasks show that the proposed Loss improves performance and robustness in models like ViT and ResNets across classification and segmentation tasks.

**Strengths:**

The proposed LJ potential seems to provide a novel approach to feature clustering. Empirical validation shows enhanced performance across several architectures and datasets.

**Weaknesses:**

1. Writing: Personably, I think that this submission is tedious and verbose. In the main body, many parts, such as Sec. 3.1 and 3.2, can be removed to save space for the content in the appendix, such as more experiments. Also, the logic in the appendix is unclear to me where often I do not understand why some descriptions should appear. I strongly suggest the authors to improve the writing.

2. Terms: I do not understand why the proposed loss is "self-supervised". Also, what is "RE-representation"? This term appears 3 times only, including one in the title, one in the abstract, and the one in the conclusion. What exactly does it mean????

3. Experiments: In current format (main body + appendix), I am not convinced by the results. Ablation study is missing. The improvements are marginal. There is no state-of-the-art comparison, or comparison with other additive losses, or simply regularizations. I do not see the value of adding such a loss.

**Questions:**

See my comments in Weaknesses.

---

### Official Review · Reviewer_A8HJ · 2024-11-02

**Soundness:** 3
**Presentation:** 3
**Contribution:** 2
**Rating:** 5
**Confidence:** 3

**Summary:**

This paper introduces the Lennard-Jones (LJ) potential from thermodynamics to the field of self-supervised re-representation. In particular, a Lennard-Jones is proposed to regularize the downstream task training.
The proposed loss is motivated from physics with repulsive force and attractive force interaction in the design.
Theory from physics gives some explanations to overfitting issue.

Experiments are conducted on 2D image recognition, and 3D point cloud classification and segmentation tasks, with ViT as the backbone.
From the results, introducing LJ loss increases the performance of various tasks.

**Strengths:**

### A novel method from thermodynamics
- This method is well-motivated from physics and thermodynamics. The replusive and attractive forces align with the self-supervised pair interactions.
- Discussion with contrastive learning method is presented.
- Theory of physics try to explain the overfitting issue from a new perspective.

### LJLoss outperforms No LJLoss in most settings
- Comparing the introduction of LJLoss and No LJLoss, the performance gains are consistent and large.
- Both 2D and 3D tasks (classificaiton and segmentation) benefit from the LJLoss.

**Weaknesses:**

### What is the connection between LJLoss and the concept of Global and local features?
- Sec. 3.2 desribes global and local features in detail. But it is unclear or there is no detailed explanation about the relationship with LJLoss

### Experimental verification is only LJLoss and NoLJLoss
- The experiments verify the effectiveness of LJLoss as an effective regularizer on CE loss of downstream finetuning tasks.
However, how about the Contrastive loss? The paper claims the benefits of LJ over contrastive.
- As a regularization loss, regularization term weight $\lambda$ is an important hyper-parameter. I did not find ablation on this term.
- Same to other hyper-parameters such as $\sigma$ and $\epsilon$.

### Other minor issues
- What does SE Loss in Figure 1 mean?
- Typos such as left and right quotations, e.g. line 081, line 125, and line 126.

**Questions:**

- What is the relationship betweem Global/Local features and the LJLoss?
- Is there any numerical comparison between the contrastive loss and LJLoss?
- Hyper-parameter effect of $\lambda$, $\sigma$, and $\epsilon$.
- What does SE Loss in Figure 1 mean?

---

### Official Review · Reviewer_h8v8 · 2024-11-03

**Soundness:** 2
**Presentation:** 3
**Contribution:** 2
**Rating:** 3
**Confidence:** 5

**Summary:**

This paper proposes a new loss function called LJ loss for self-supervised learning. LJ loss can automatically constrain similar sample features to attract each other and dissimilar samples to stay away from each other. Experimental results show the effectiveness of the proposed method.

**Strengths:**

If I have to say the highlight of this article, it is that it proposes a new loss function based on the lennard-jones potential, which automatically constrains similar samples to be close to each other and dissimilar samples to be separated from each other.

**Weaknesses:**

1. The motivation for this article can be summarized as the search for a means by which similar samples can be automatically constrained to attract each other and dissimilar samples to stay away from each other. Similar solutions have been proposed by researchers long ago, e.g., in the literature [1]. However, this article does not analyze it comparatively.

[1] Song, Zeen, et al. "On the Discriminability of Self-Supervised Representation Learning." arXiv preprint arXiv:2407.13541.

2. The related work section is pretty weak. First, the authors ignore very recent hot network architectures such as transformer and KAN. Second, the authors have a one-sided understanding of self-supervised learning. There are many self-supervised methods that do not require positive and negative samples, such as BYOL, Barlow Twins, and MAE. Finally, this understanding of physically-guided deep learning is also inadequate. For example, the authors do not mention ReduNet, a learning framework based on the Principle of Maximizing Rate Reduction.

3. Section THEORY is over-claimed. This chapter deals with the general form and characterization of the Lennard-Jones potential and contains no new concepts, definitions, and theorems. At the same time, the authors do not give the logical relationship between Section 3.2 and Section 3.3, and it seems that Section 3.2 is redundant.

4. As shown in SimCLR, MoCo, BYOL, and Barlow Twins, we can see that self-supervised learning not only performs well on tasks similar to the training data, but also on transfer tasks. There are often significant differences between different tasks, and more mining of valid information in the training task may lead to overfitting of the training task, thus affecting the transferability of the self-supervised learning method. Does the LJ loss proposed in this paper reduce feature mobility? If not, please conduct the corresponding theoretical analysis.

**Questions:**

1. In Lines 288-293 is confusing. It is always known that for classification tasks, the larger the distance between classes, the more helpful it is for classification, and this idea is also the core idea of SVM. But the authors go on to emphasize that classification problems should focus only on intra-class distances.

2. The authors repeatedly emphasize the problems in self-supervised learning, but the experimental design does not include a comparison with self-supervised learning methods, which is puzzling.

---

### Official Review · Reviewer_a6Wg · 2024-11-11

**Soundness:** 2
**Presentation:** 2
**Contribution:** 2
**Rating:** 3
**Confidence:** 4

**Summary:**

Inspired by the Lennard-Jones potential, this work proposed the corresponding loss function to help computer vision tasks. By setting hyperparameters appropriately, the loss aims to balance intra-class and inter-class distributions. Experiments are conducted on multiple tasks for evaluation.

**Strengths:**

1. Exploring theoretical results from physics for deep learning is an interesting direction.

2. This work considers multiple computer vison tasks, e.g., classification, segmentation, etc.

3. Both ViT and ResNet are adopted in evaluation to demonstrate the effectiveness of the proposed loss function.

**Weaknesses:**

1. The motivation is unconvincing. In L247, it shows that the system is with zero net force when $r=\sigma$. However, the gradient does not equal to zero in that case and the value of the loss function can be smaller than 0 by minimizing it. Moreover, the analysis below Enq.7 is inconsistent with that below Enq.5. For example, when $r<\sigma$, which means the pair of examples are similar, the analysis for Eqn.5 shows that the repulsive term dominates. On the contrary, that for Eqn.7 states that the attractive forces dominate. According to my understanding, the loss function in Eqn.6 just pushes all pairs to a pre-defined similarity $\sigma$, which is more close to the statement after Eqn. 5 and it cannot help representation learning significantly.

2. The data sets for experiments are quite limited. Note that only small data sets, e.g., CIFAR and TinyImageNet, are applied for classification. Lager benchmark data sets, e.g., ImageNet, should be included for comparison.

3. The effectiveness of the proposed method is not well justified. The performance of baseline with the proposed loss function is even worse than the original baseline as illustrated in Fig. 5.

4. The parameter $\sigma$ is crucial for the success of the loss function and can be sensitive for different tasks. Including an ablation study on parameters can be better for demonstration the proposed method.

**Questions:**

Please kindly check the above weaknesses.

---

### Note · Authors · 2024-11-19

**Comment:**

Thank you for the reviewers' professional comments. The theoretical foundation of this work is not solid, and the experiments are insufficient. Therefore, we need to withdraw the manuscript. We apologize for any inconvenience caused.

**Withdrawal Confirmation:**

I have read and agree with the venue's withdrawal policy on behalf of myself and my co-authors.